

# RNA sequencing analysis of FGF2-responsive transcriptome in skin fibroblasts

Baojin Wu[1,*], Xinjie Tang[1,*], Zhaoping Zhou[1], Honglin Ke[2], Shao Tang[3] and Ronghu Ke[1]

[1] Department of Plastic Surgery, Huashan Hospital Affiliated to Fudan University, Shanghai, China
[2] Department of Emergency, Huashan Hospital Affiliated to Fudan University, Shanghai, China
[3] Department of Statistics, Florida State University, Tallahassee, FL, USA
[*] These authors contributed equally to this work.

## ABSTRACT

**Background.** Fibroblast growth factor 2 (FGF2) is a highly pleiotropic cytokine with antifibrotic activity in wound healing. During the process of wound healing and fibrosis, fibroblasts are the key players. Although accumulating evidence has suggested the antagonistic effects of FGF2 in the activation process of fibroblasts, the mechanisms by which FGF2 hinders the fibroblast activation remains incompletely understood. This study aimed to identify the key genes and their regulatory networks in skin fibroblasts treated with FGF2.

**Methods.** RNA-seq was performed to identify the differentially expressed mRNA (DEGs) and lncRNA between FGF2-treated fibroblasts and control. DEGs were analyzed by Gene Ontology (GO) and Kyoto Encyclopedia of Genes and Genomes (KEGG). Furthermore, the networks between mRNAs and lncRNAs were constructed by Pearson correlation analysis and the networkanalyst website. Finally, hub genes were validated by real time-PCR.

**Results.** Between FGF2-treated fibroblasts and control fibroblasts, a total of 1475 DEGs was obtained. These DEGs were mainly enriched in functions such as the ECM organization, cell adhesion, and cell migration. They were mainly involved in ECM-receptor interaction, PI3K-Akt signaling, and the Hippo pathway. The hub DEGs included COL3A1, COL4A1, LOX, PDGFA, TGFBI, and ITGA10. Subsequent real-time PCR, as well as bioinformatics analysis, consistently demonstrated that the expression of ITGA10 was significantly upregulated while the other five DEGs (COL3A1, COL4A1, LOX, PDGFA, TGFBI) were downregulated in FGF2-treated fibroblasts. Meanwhile, 213 differentially expressed lncRNAs were identified and three key lncRNAs (HOXA-AS2, H19, and SNHG8) were highlighted in FGF2-treated fibroblasts.

**Conclusion.** The current study comprehensively analyzed the FGF2-responsive transcriptional profile and provided candidate mechanisms that may account for FGF2-mediated wound healing.

Corresponding author
Ronghu Ke, ronghuke@163.com

## INTRODUCTION

Fibrosis is a fundamental wound healing process that occurs in almost every organ including lung, liver, kidney, heart, or skin. Despite numerous crucial differences among fibrotic pathologies of various organs, one of the commonalities among these affected organs is the activation and transdifferentiation of quiescent fibroblasts into contractile myofibroblasts (*Yazdani, Bansal & Prakash, 2017*). During the process of fibrosis, fibroblasts and myofibroblasts are the main effectors. Compared to non-activated fibroblasts, myofibroblasts are larger in the area and have their structural characterization with the presence of actin filament bundles containing alpha-smooth muscle actin ($\alpha$-SMA), which augments its ability to generate contractile force in the wound site. Myofibroblasts are also characterized as producing excessive extracellular matrix (ECM), particularly type I collagen and the fibronectin-extra domain A (EDA) isoform. Soon after their injury, local fibroblasts translate into the core of the wound and differentiate into contractile myofibroblasts, leading to wound retraction and healing. After closing the wound, myofibroblasts undergo apoptosis to clear the wound site (*Vallee & Lecarpentier, 2019*). However, under pathological conditions, persistent myofibroblasts activation results in an overproduction of collagen and the formation of the pathological scar. Thus, fibroblast differentiation into myofibroblasts is the key process in dysregulated wound healing as well as organ fibrosis. Intervening into myofibroblast-induced pro-fibrotic activities using drug targeting technologies can be a promising approach for developing novel therapeutics against fibrosis (*Yazdani, Bansal & Prakash, 2017*).

Fibroblast growth factor 2 (FGF2), also known as a basic fibroblast growth factor (bFGF), is one of the family members of mammalian fibroblast growth factors. FGF2 has low (18-kDa) and high (22-, 22.5-, 24-, and 34-kDa) molecular weight isoforms, which are translated from a single transcript by starting from alternative, in-frame start codons. These isoforms act predominantly in an autocrine or paracrine manner via the fibroblast growth factor receptors (FGFRs), which contains two receptor isoforms (IIIb or IIIc). FGF2 preferentially activates FGFR1c, FGFR3c and FGFR4 and shows some affinity to FGFR1b and FGFR2c (*Ornitz et al., 1996*). When FGF2 activates its receptor, intracellular adaptor and effectors proteins are recruited to stimulate the signal pathways, most notably the mitogen-activated protein kinases (MAPKs) and Akt/mTOR. By the canonical pathways, FGF2 was well studied especially in the field of ontogenesis, stem cell self-renewal, and tissue repair (*Akl et al., 2016*). As for FGF2 roles in tissue repair, enormous clinical applications particularly in China and to an extent in Japan, have been carried out. Based on the clinical research, FGF2 has been shown to have anti-fibrotic effects in conditions as diverse as burns, chronic wounds, oral ulcers, vascular ulcers, diabetic ulcers, pressure ulcers, and surgical incisions (*Nunes et al., 2016*; *Akita et al., 2008*; *Matsumine, 2015*; *Ono et al., 2007*). Moreover, FGF2 antagonized TGFβ1-induced differentiation of fibroblasts and thus affected fibrosis during wound repair (*Dolivo, Larson & Dominko, 2017a*). Importantly, FGF2 was observed to induce a shift in gene expression to a more anti-fibrotic signature attenuated the expression of pro-fibrotic genes, including collagen I, collagen III, and $\alpha$-SMA (*Dolivo, Larson & Dominko, 2017b*).

Despite the extensive observations, the mechanisms by which FGF2 regulates the fibrotic response remain incompletely understood.

Microarray and high-throughput sequencing technologies are powerful tools that can be used to investigate potential target genes for diseases and underlying pathological mechanisms (*Mery et al., 2019*). By microarray technology, the expression profiles have been investigated in FGF2-treated fibroblasts (*Hernandez & Dominko, 2016*; *Kashpur et al., 2013*). In these studies, high throughput transcriptional datasets were acquired to decipher the significant genes and pathways in FGF2-treated fibroblasts. Compared to microarrays, RNA-Seq technology shows higher sensitivity and increased quantitative accuracy and therefore detects even low abundance transcripts. Additionally, the RNA-Seq expands in terms of non-coding RNA detection (*Schwingen, Kaplan & Kurschus, 2020*). Here, we performed RNA-seq analysis for the dissection of the molecular profiles of FGF2-treated fibroblasts. Following the screening out the differentially expressed genes (DEGs), we identified the key genes and the signaling pathways triggered by FGF2 in skin fibroblasts. Thus, the study would provide a comprehensive understanding of the mechanisms regulated by FGF2 in skin fibroblasts, which may guide subsequent studies on skin wounds.

## MATERIALS AND METHODS

### Cell cultures and reagents

Human skin biopsies were obtained from healthy subjects (File S1). All the tissues were collected in Dulbecco's modified Eagle's medium (DMEM; Gibco BRL, Grand Island, NY, USA), and primary fibroblasts were established as described previously (*Xuan et al., 2014*). Primary fibroblasts were grown in DMEM supplemented with 10% fetal bovine serum (PAA Laboratories, Etobicoke, Ontario, Canada), 100 U/mL penicillin, and 100 µg/mL streptomycin (Gibco BRL) at 37 °C in a humidified atmosphere containing 5% $CO_2$. When confluent, the cells were trypsinized using a 0.25% trypsin/0.02% EDTA solution (Sigma, St Louis, MO, USA) and subcultured at a 1:3 ratio on culture plastic dishes. After three or six passages, fibroblasts were used for the following experiments. The human skin fibroblasts were cultured in serum-free DMEM alone or DMEM with FGF2 (Proteintech, Rosemont, IL, Cat: HZ-1285) at varying doses (0, 5, 10, and 50 ng/ml) for 48 hrs. For RNA-seq assay, cells were treated with 10ng/ml FGF2.

### RNA isolation and RNA sequencing

After 48 h of culture, total RNA was extracted from seven samples (four samples from FGF2-treated fibroblasts, three samples from control fibroblasts) using Trizol (Invitrogen, Carlsbad, CA). The libraries were constructed by RNA Library Prep Kit for Illumina (NEB, MA, and USA) according to the manufacturer's instructions. The library products were sequenced using Illumina HiSeqTM 2500 (Illumina, CA, USA). Index of the reference genome was built and paired-end clean reads were aligned to the reference genome using STAR. HTSeq v0.6.0 was used to count the reads numbers mapped to each gene. Then, the FPKM of each gene was calculated based on the length of the gene and reads count mapped to this gene. Differentially expressed RNA between the two groups was performed using the

DESeq. 2R package (1.10.1). The resulting $p$ values were adjusted using the Benjamini and Hochberg's approach. Genes with an adjusted $p < 0.05$ and $|log2FC|>1.0$ were assigned as differentially expressed genes (DEGs) or LncRNAs.

## Annotation for DEGs

The DAVID Functional Annotation Bioinformatics Microarray Analysis (website: https://david.ncifcrf.gov/) was used for the GO enrichment and KEGG pathway analysis (*Xu et al., 2020*). The GO Annotation for DEGs included three terms, molecular functions (MF), biological processes (BP), and cellular components (CC) of genomic products. As for pathway enrichment, DEGs were analyzed by the Kyoto encyclopedia of genes and genomes (KEGG) pathway within the DAVID database (Version 6.7). $P < 0.05$ was considered as statistical significance for the correlations. The Bubble Charts were conducted using the ggplot2 package in R software.

## LncRNA-mRNA co-expression network construction

To screen the significant lncRNAs, the LncRNA-TF network was constructed by the NetworkAnalyst website (https://www.networkanalyst.ca/). The parameters for the enrichment analysis were as follows. A specific organism was chosen H. sapiens (human). The ID type was chosen official gene symbol. TF-gene interaction was analyzed using the ENCODE database (*Wang et al., 2020a*). Based on the function of ECM organization in wound healing, the differentially expressed mRNAs involved in ECM organization were focused to analyze the associations between mRNAs and lncRNAs. The correlations of mRNAs and lncRNAs were calculated by the Pearson coefficients to construct the networks. The significant pairs of mRNAs-lncRNAs (coefficient > 0.95, $P < 0.05$) were visualized by Cytoscape software.

## Quantitative real-time PCR

Total RNA was isolated from tissues or cells by using TRIZOL reagent (Invitrogen, Carlsbad, CA) according to the manufacturer's protocol. First strand cDNA synthesis was performed on the total RNA (0.5 μg) using the reverse transcription kit (Takara, Dalian, China). The real-time PCR assay was conducted with the SYBR Green PCR Kit (Takara, Dalian, China). The primers for target genes are listed in File S2. The PCR reaction conditions were as follows: 95 °C for 15 s followed by 40 cycles of 95 °C for 5 s, and 60 °C for 30 s. Real-time PCR results were performed by the $\Delta\Delta$CT method.

# RESULTS

## Transcriptional profiles in FGF2-treated fibroblasts

To determine the effects of FGF2 on the fibroblasts, skin fibroblasts were treated with FGF2 at different doses for 48 h. As shown in Fig. 1A, morphological results from the microscope demonstrated skin fibroblasts revealed a smaller size, less spindle-like shape in a dose-dependent manner, indicating fibroblasts activation was suppressed by FGF2. FGF2 treatment at 10 ng/ml or 50 ng/ml significantly induced morphological changes in fibroblasts. Based on the FGF2 concentration (4 ng/ml) used in previous microarray

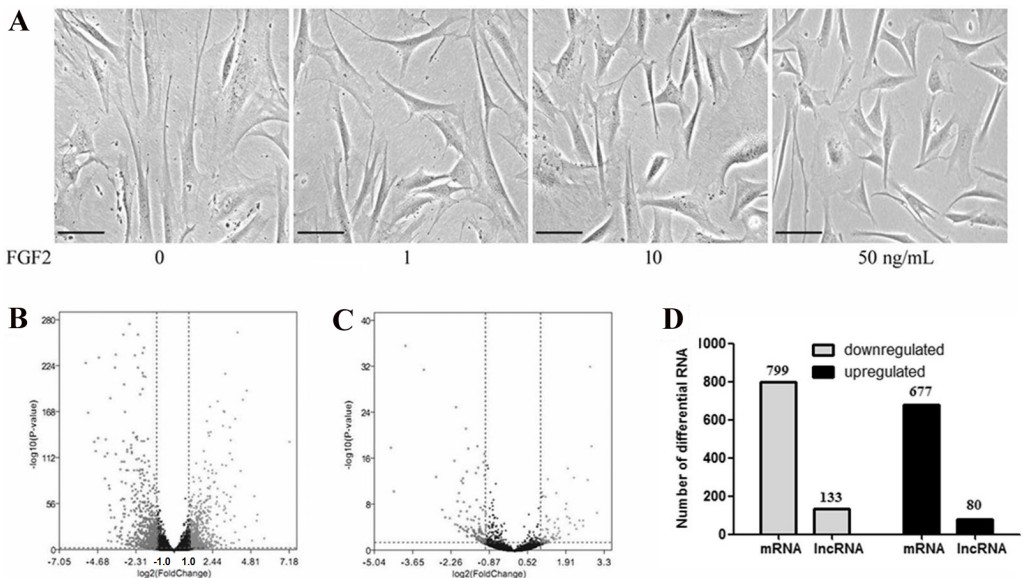

**Figure 1** **Transcriptional profiles in skin fibroblasts treated with FGF2.** Transcriptional profiles in skin fibroblasts exposure to FGF2. (A) Representative morphology of skin fibroblasts treated with FGF2 at indicated dose. (B) The volcano plot of mRNAs expression profiles. (C) The volcano plot of LncRNAs expression profiles. $P < 0.05$ and fold change <2. (D) Summary of differently expressed mRNAs and LncRNAs.

studies (*Hernandez & Dominko, 2016*; *Kashpur et al., 2013*), skin fibroblasts were treated with FGF2 at 10 ng/ml for 48 h and harvested for RNA-seq to understand the expression profiles in FGF2-treated fibroblasts. The RNA-seq result showed a total of 1475 mRNAs were differentially expressed in FGF2-treated fibroblasts (Fig. 1B, File S3), among which 676 genes were up-regulated and 799 genes were down-regulated (Fig. 1C). Meanwhile, there were 213 LncRNAs with |fold change (FC)|>2 (Fig. 1B, File S4), among which 80 LncRNAs were up-regulated and 133 were down-regulated (Fig. 1C).

## GO enrichment for DEGs

To explore the functions of the DEGs, the 1475 DEGs were annotated by three GO categories (biological processes, molecular functions, and cellular components). For GO biological process, DEGs were involved in the extracellular matrix (ECM) organization, angiogenesis, cell adhesion, positive regulation of cell migration, and collagen catabolic process (Fig. 2, File S5). For molecular function, the DEGs were significantly enriched in the regulation of calcium ion binding, growth factor activity, ECM structural constituent, heparin binding, and frizzled binding (Fig. 2). For cellular component annotation, the most significant terms were enriched in extracellular space, extracellular region, proteinaceous ECM, ECM, and plasma membrane (Fig. 2).

## KEGG pathway analysis

The KEGG pathway analysis was performed to identify the significant pathways induced by FGF2 in human skin fibroblasts. The 1475 DEGs were mapped to 44 KEGG pathways.

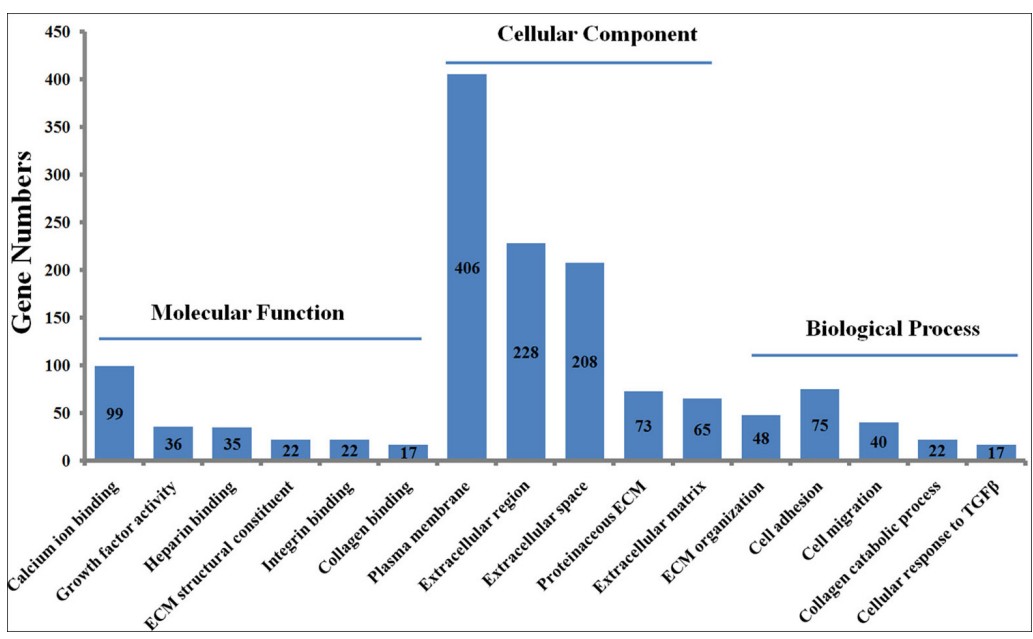

**Figure 2** **GO enrichment analysis of differently expressed genes (DEGs) in skin fibroblasts treated with FGF2.** The bar chart visualizes the top five significantly enriched items that are grouped by molecular functions, cellular component and biological process. The x-axis indicates the term of GO enrichment, and Y-axis represents the number of hub genes enriched in a certain term.

Among these pathways, 37 pathways were significantly enriched ($P \leq 0.05$) (File S6). The top 10 significant pathways were represented, including ECM-receptor interaction, PI3K-Akt signaling pathway, Hippo signaling pathway, complement and coagulation cascades, TGF-beta signaling pathway, MAPK signaling pathway, proteoglycans in cancer, protein digestion, focal adhesion, and cell adhesion molecules (Fig. 3).

## mRNA-lncRNA co-expression network

To investigate the regulatory networks of lncRNAs, the differentially expressed lncRNAs were uploaded within the NetworkAnalyst website. The PPI networks of 213 lncRNAs revealed the top 10 significant LncRNAs, including HOXA-AS2, LOC100130417, H19, LOC100507420, and SNHG8 (Fig. 4A). Based on the above result of GO enrichment, ECM organization was significantly involved in the FGF2-mediated biological process. There were 48 differentially expressed mRNAs involved in ECM organization. To explore associations between the 48 mRNAs and lncRNAs, pairs of mRNAs-lncRNAs were analyzed by the Pearson correlation analysis and were sorted by the coefficients, which represented the associations of mRNA and LncRNA in the networks (File S7). As shown in Fig. 4B, the three key LncRNAs, including HOXA-AS2, H19, and SNHG8 were identified in the FGF2-mediated ECM organization. The three LncRNAs were significantly associated with six DEGs, including LOX, TGFB1, ITGA10, COL4A1, COL3A1, and PDGFA. Among the 6 DEGs, only ITGA10 was upregulated and the other 5 DEGs were downregulated (Fig. 4B).

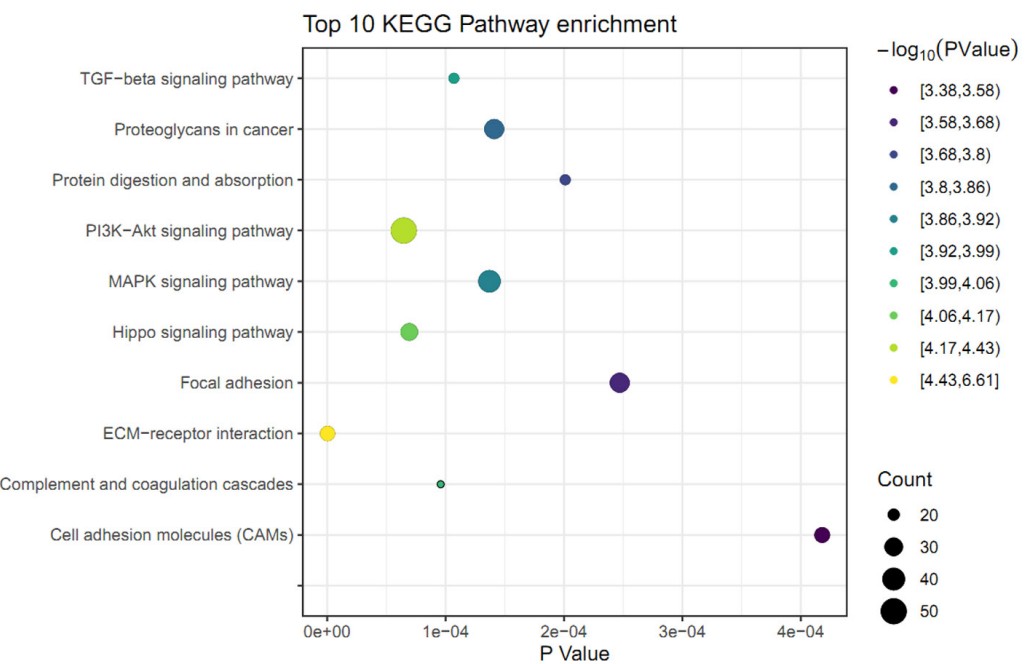

**Figure 3 KEGG pathway analysis of differently expressed genes (DEGs) in skin fibroblasts treated with FGF2.** The bubble diagram represents the top 10 significantly enriched pathway. The *x* and *y* axis indicate the *P* value and pathways, respectively. Dot size denotes the count of enriched genes and dot color indicates *P* value.

## Validation of hub genes by real-time PCR

To further verify the results of RNA-seq analysis, the levels of the six hub genes (COL3A1, COL4A1, LOX, PDGFA, TGFBI, and ITGA10) were validated. Firstly, the levels of six hub genes were compared between two GEO datasets (GSE60580 and GSE48967) and our RNA-seq data. As illustrated in Fig. 5A, our RNA sequencing data and the GEO datasets demonstrated ITGA10 was significantly upregulated while the other five genes were downregulated in FGF2-treated fibroblasts. Next, the levels of six hub genes were validated by real-time PCR. As predicted by the above bioinformatics analysis, real-time PCR results also showed ITGA10 was significantly upregulated while the other five genes were downregulated in FGF2-treated fibroblasts (Fig. 5B).

## DISCUSSION

Although FGF2 has been revealed to have the potential to attenuate fibrotic phenotypes and drive the more desirable, regenerative resolution of wound healing in damaged tissue, the mechanisms by which FGF2 ameliorates fibrosis are not entirely understood. As mentioned above, fibroblasts and myofibroblasts are the main effectors in the process of fibrosis. Of note, in vitro fibroblasts grown on two-dimensional (2-D) substrata flatten out and develop prominent stress fibers that are made of actin, myosin, and actin-binding proteins. This is in marked contrast with the in vivo situation, in which actively migrating fibroblasts do not contain stress fibers (*Walpita & Hay, 2002*). Thus, in vitro fibroblasts grown on tissue

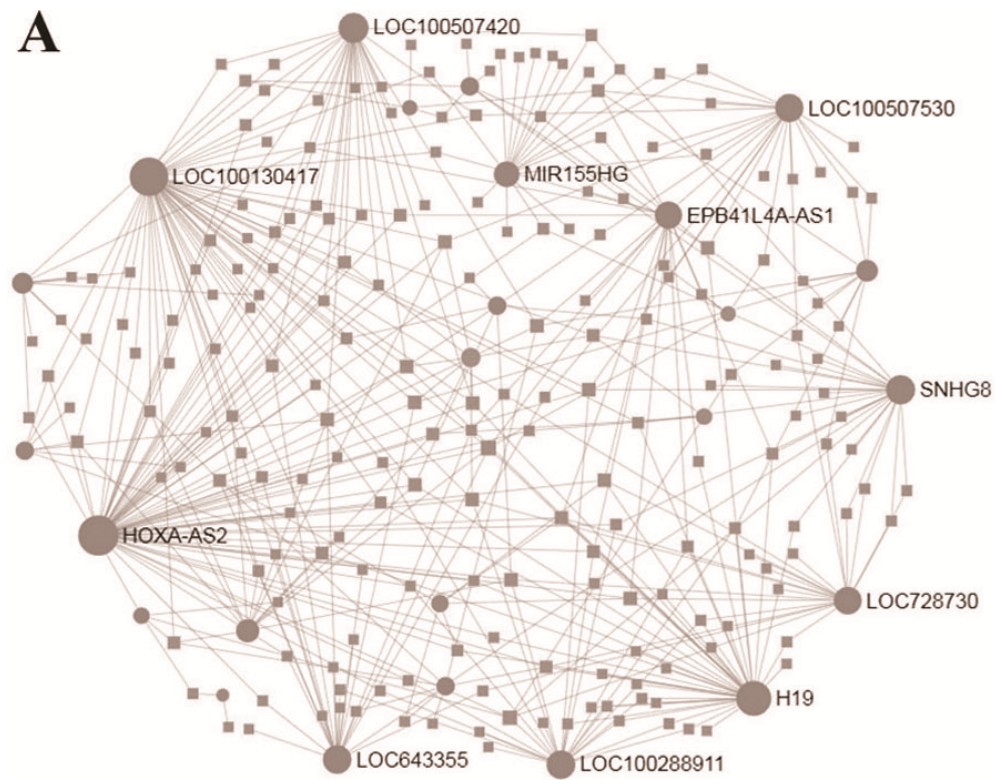

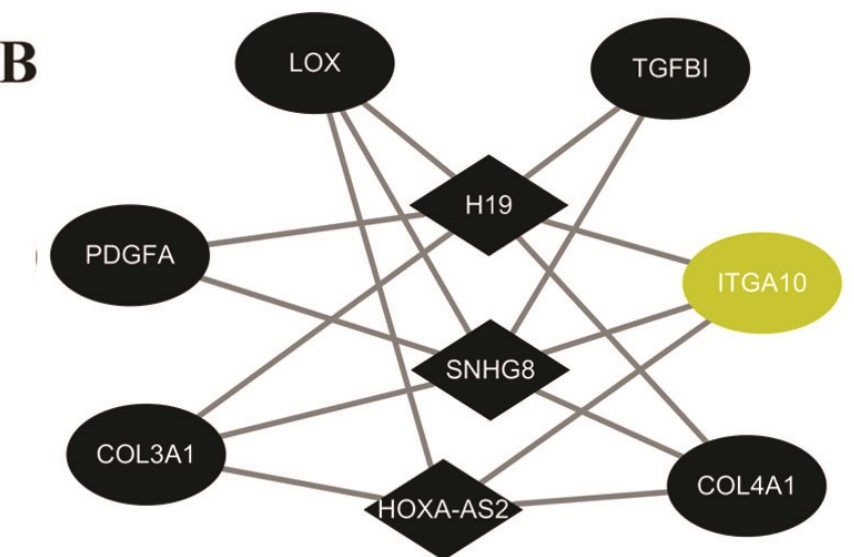

**Figure 4 Networks of protein-protein interaction.** (A) Networks of the differently expressed LncRNAs created by the NetworkAnalyst website. (B) The co-expression network of the ECM-associated mRNAs-lncRNAs created by Pearson's correlation. The diamond and ellipses nodes denote significant LncRNAs and mRNAs, respectively. The yellow and black nodes indicate upregulated and downregulated transcripts, respectively.

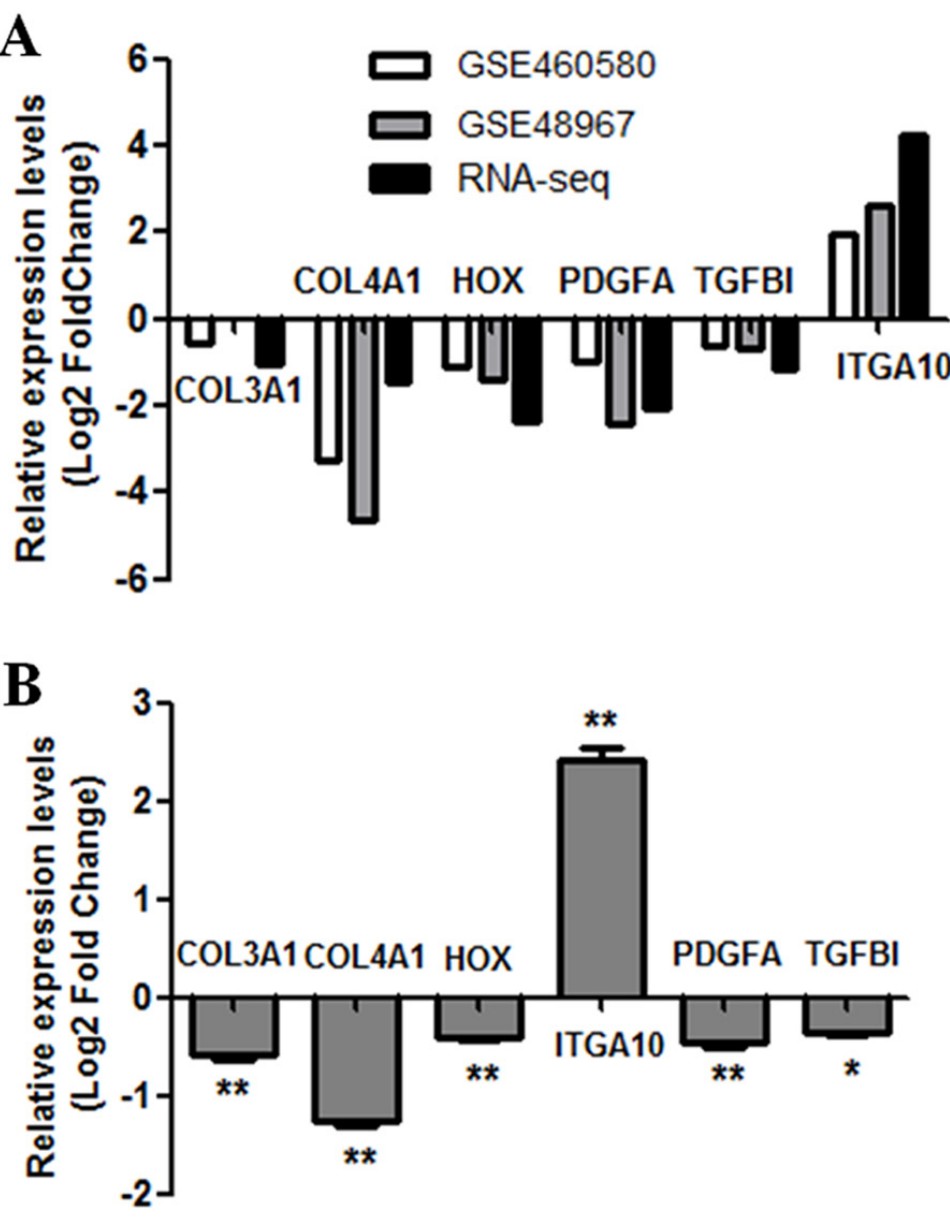

**Figure 5** **Validation of the expression for select genes.** (A) The expression levels of seven genes in previous GEO microarray datasets and the present RNA-seq data. (B) The validation of 7 DEGs by real-time PCR. The x-axis shows DEGs and y-axis shows log2(FoldChange). The log2(FoldChange)>0 and log2(FoldChange) <0 indicate upregulation and downregulation, respectively. Statistical significance was assessed by Student's t-test. $*P < 0.05$, $**P < 0.01$.

culture plastic or glass tend to reveal myofibroblast phenotypes. Consistent with the notion that in vitro fibroblasts are subject to be high stiffness, our study showed myofibroblast markers including ACTA2 were expressed in fibroblasts grown on plastic dishes. And the effects of FGF2 on transcriptomics of fibroblasts in the study may be more representative of the effects of pharmacologic exogenous FGF2 on myofibroblasts compared to the effects of

endogenous FGF2 on quiescent fibroblasts. By the RNA-seq analysis, our study identified 1475 FGF2-responsive DEGs, including LOX, TGFB1, PDGFA, COL3A1, COL4A1, and ITGA10. Similar to our result, previous microarray assays were performed to identify the expression profile of FGF2-treated fibroblasts (*Hernandez & Dominko, 2016*; *Kashpur et al., 2013*). Based on the advantages of the RNA-Seq in terms of lncRNAs detection, our results further suggested HOXA-AS2, H19, and SNHG8 were involved in FGF2-mediated ECM organization. Therefore, the study would provide a comprehensive understanding of the FGF2-responsive genes in human skin fibroblasts, which may guide subsequent studies on wound healing.

Our first goal was to identify significantly differentially expressed genes (DEGs) in FGF2-treated fibroblasts. Fibroblasts are responsible for ECM production during dermal wound healing. Here, our results showed that exogenous FGF2 affects a large number of genes involved in the production and remodeling of ECM. Type III collagen is a hallmark of several chronic fibrotic diseases including systemic sclerosis, cardiac fibrosis, lung fibrosis, liver cirrhosis, and renal fibrosis (*Kuivaniemi & Tromp, 2019*). Besides collagen III, FGF2 caused downregulation of other collagens such as collagen I, collagen V, collagen IV, collagen XI, and collagen XV, as well as caused upregulation of collagen X, collagen XIII, and collagen XVII (File S3). The subsequent qRT-PCR analysis confirmed the downregulation of COL3A1 and COL4A1 in FGF2-treated fibroblasts (Fig. 5B). In line with our result, the previous microarray study demonstrated COL3A1 and COL4A1 were downregulated by FGF2 in dermal fibroblasts (*Kashpur et al., 2013*). Besides the collagen, our result revealed other ECM genes affected by FGF2 treatment including laminins and fibronectins (File S3). Similarly, FGF2 has previously been shown to significantly downregulated laminin alpha 2 (LAMA2) and Fibronectin 1 (FN1) (*Kashpur et al., 2013*). Thus, our study, together with previous observations, indicated FGF2 modulated the production of the ECM in human fibroblasts, which potentially favored the changes in cell attachment to ECM. Cell attachment to the ECM is regulated through integrins. Integrins constitute a subset of the integrin family with the affinity for GFOGER-like sequences in collagens and are crucial for dynamic connective tissue remodeling events–such as wound healing (*Zeltz & Gullberg, 2016*). Here, our results demonstrated FGF2 caused upregulation of numerous integrins such as integrin alpha 10 (ITGA10), integrin alpha 2, and integrin alpha 6, as well as caused downregulation of integrin beta like 1, integrin beta 8, and integrin beta 4 (File S3). Most profoundly affected by FGF2 treatment was ITGA10. The qRT-PCR analysis confirmed the upregulation of ITGA10 (Fig. 5B). ITGA10 was originally identified as a type II collagen-binding receptor on chondrocytes and mainly confined to cartilage-containing tissues (*Camper et al., 2001*). ITGA10 was also observed to b up-regulated in malignant melanoma cells. And further investigation showed downregulating ITGA10 expression by an inhibitory antibody or an antisense construct had hindered migratory potential, suggesting a role for ITGA10 in melanoma cell migration (*Wenke et al., 2007*). Here, our study showed that exogenous FGF2 significantly induced the expression of ITGA10. Consistent with our result, the induce of ITGA10 expression by FGF2 previously occurred in mesenchymal stem cells and dermal fibroblasts (*Kashpur et al., 2013*; *Varas et al., 2007*).

Thus, our study, together with previous observations, suggested FGF2 modulated the ITGA10 expression in human fibroblasts, which favored the migratory potential.

Following the identification of DEGs, GO analysis was performed to identify the important biological processes in human FGF2-treated fibroblasts. Our GO analysis demonstrated that the FGF2-associated DEGs were mainly enriched in the ECM organization, cell adhesion, and cell migration. This result accorded with the knowledge that FGF2 functioned as an important regulator in cell behavior, cell growth, and survival (*Akl et al., 2016*; *Klagsbrun, 1992*; *Przybylski, 2009*). Also, we further explored the effect of FGF2 on signal pathways in skin fibroblasts. KEGG enrichment results implied that FGF2 was mainly involved in classical pathways including ECM-receptor interaction and PI3K-Akt signaling pathway. Cell interactions with the ECM are mediated by integrins and various signaling cascades are activated, which control cell adhesion, proliferation, morphogenesis, differentiation, and survival (*DiPersio & Van De Water, 2019*). Of note, our enrichment results indicated that FGF2 was significantly associated with the Hippo signaling pathway. In line with our results, FGF2 was observed to promote the Hippo/YAP-signaling by inducing the nuclear-YAP expression during lens cell proliferation and differentiation, indicating FGF2 plays important roles in mediating the Hippo signaling pathway (*Dawes et al., 2018*). Furthermore, FGFR1 and FGFR2 were showed to directly interact with YAP/TAZ at multiple tyrosine residues independent of upstream Hippo signaling (*Azad et al., 2020*). Thus, our study, together with previous observations, suggested that the pivotal role of the FGF/FGFR signaling in mediating the Hippo signaling pathway.

In addition to coding genes, this present study was focused on the differentially expressed LncRNAs in FGF2-treated fibroblasts. LncRNAs have been linked to the biological processes in various skin cells both physiological and pathological conditions, Although the role of lncRNAs in normal skin wound healing remains unexplored, emerging observations have linked lncRNAs to pathological scars. By microarray analysis, more than 2,500 lncRNAs were differentially expressed in keloid tissue compared with the normal human skin (*Liang et al., 2015*). Here, our study has revealed 213 differentially expressed LncRNAs and highlighted the three key LncRNAs (HOXA-AS2, H19, and SNHG8) in FGF2-treated fibroblasts. LncRNA H19, as a 2.3 kb lncRNA, is encoded from paternally imprinted and maternally expressed on human chromosome 11p15.5. The imprinted H19 is highly expressed in embryogenesis but is barely detectable in most tissues shortly after birth (*Lustig et al., 1994*). Numerous studies have revealed aberrant alterations of H19 expression in various tumors, implicating a crucial role of H19 in tumorgenesis (*Ghafouri-Fard, Esmaeili & Taheri, 2020*). Recently, H19 has also been observed to be upregulated in keloid tissues and fibroblasts. Moreover, silencing of H19 promoted cell viability, migration, and invasion of the fibroblasts (*Wang et al., 2020b*). In the present study, H19 was downregulated in the dermal fibroblast exposure to FGF2, suggesting FGF2 attenuated the expression of H19. Conversely, *Sun et al. (2019)* reported the H19 levels were remarkably increased in FGF2-treated human umbilical vein endothelial cells. Thus, our study indicated the FGF2-mediated H19 expression appeared to exhibit a context-dependent pattern. In addition to H19, another two LncRNA HOXA-AS2 and SNHG8 were identified in FGF2-treated fibroblasts. Although numerous observations have demonstrated HOXA-AS2 and

SNHG8 play vital roles in the development of various cancer including non-small cell lung cancer and breast cancer, gastric cancer (*Chen et al., 2018*; *Wang et al., 2018*). But, whether HOXA-AS2 and SNHG8 exhibit a certain function in fibroblasts remains elusive. Here, our result showed HOXA-AS2 and SNHG8 were downregulated in FGF2-treated fibroblasts. Taking into account the significance of FGF2 in skin wound healing, we conjectured that the novel lncRNAs (H19, HOXA-AS2. and SNHG8) may provide candidate mechanisms that may account for FGF2-mediated wound healing.

## CONCLUSIONS

In summary, the current study carried out the RNA-seq analysis to identify the crucial genes in FGF2-treated skin fibroblasts. Our results showed FGF2 was associated with ECM organization as well as other biological processes including cell adhesion and cell migration. Furthermore, our study identified the key genes (LOX, TGFB1, PDGFA, COL3A1, COL4A1, and ITGA10), with ITGA10 being particularly prominent. Notably, our study highlighted the three key lncRNAs (HOXA-AS2, H19, and SNHG8) in FGF2-treated fibroblasts. Further studies are needed to delineate the mechanism that underlies the key LncRNAs in FGF2-mediated cellular functions. Therefore, the present study may provide new ideas and targets for the diagnosis and treatment of skin wound healing.

## ACKNOWLEDGEMENTS

The authors gratefully thank Zi-Qi Zhou for his statistical assistance.

### Funding
This work was supported by funds from the National Natural Science Foundation of China (No.81401616) and the Natural Science Foundation of Shanghai (14ZR1405100). The funders had no role in study design, data collection and analysis, decision to publish, or preparation of the manuscript.

### Grant Disclosures
The following grant information was disclosed by the authors:
National Natural Science Foundation of China: 81401616.
Natural Science Foundation of Shanghai: 14ZR1405100.

### Competing Interests
The authors declare there are no competing interests.

### Author Contributions
- Baojin Wu and Xinjie Tang conceived and designed the experiments, performed the experiments, analyzed the data, prepared figures and/or tables, authored or reviewed drafts of the paper, and approved the final draft.

- Zhaoping Zhou performed the experiments, analyzed the data, prepared figures and/or tables, and approved the final draft.
- Honglin Ke and Shao Tang analyzed the data, prepared figures and/or tables, and approved the final draft.
- Ronghu Ke conceived and designed the experiments, prepared figures and/or tables, authored or reviewed drafts of the paper, and approved the final draft.

## Human Ethics

The following information was supplied relating to ethical approvals (i.e., approving body and any reference numbers):

The Ethics Committee of Huashan Hospital Affiliated to Fudan University (Ethical Application Ref: 2020-350) has approved the study.

## Data Availability

Data is available at GenBank: GSE157071.

## Supplemental Information

Supplemental information for this article can be found online at http://dx.doi.org/10.7717/peerj.10671#supplemental-information.

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
