# Peer review of "RNA sequencing analysis of FGF2-responsive transcriptome in skin fibroblasts"

_PeerJ, doi:10.7717/peerj.10671_

## Round 0.1 · original submission · Major Revisions

Your work has been reviewed by three subject experts who have raised several concerns.

Reviewer 1 ·

Basic reporting

Performed RNA-seq to explore the differentially expressed genes of skin fibroblasts in response to FGF2. Will be a good resource.

Experimental design

Good

Validity of the findings

Validation is missing. See comments below.

Additional comments

The manuscript entitled “RNA sequencing analysis of FGF2-responsive transcriptome in skin fibroblasts” is a well written manuscript. The present study performed RNA-seq to explore the differentially expressed genes of skin fibroblasts in response to FGF2. Using bioinformatics analysis, they identified 1475 FGF2-responsive differentially expressed genes, including LOX, TGFB1, 180 PDGFA, COL3A1, COL4A1 and ITGA10. However, they have not validated anyone. Although RNA-seq is a very powerful technique, some technical artifacts may be present in the data. Therefore, Validation with an independent technique is always advisable.
Here are the following major changes to be performed for improving the quality of this manuscript for publication.
1. Validation by qPCR of some of the prominent FGF2 driven differentially expressed genes in skin fibroblasts (like, LOX, TGFB1, 180 PDGFA, COL3A1, COL4A1 and ITGA10).
2. Comparison of the present FGF2 responsive genes set to the know FGF responsive gene data set. Also, a comparison of this data with genes known to be differentially regulated in wound healing could be highlighted.
3. Cellular functions of LncRNAs must be included in the discussion to highlight its possible role in wound healing.

·

Basic reporting

No comment

Experimental design

No comment

Validity of the findings

No comment

Additional comments

In this manuscript, Wu et al. seek to characterize the transcriptomic changes induced in fibroblasts by FGF2, with a focus on co-expression networks of mRNA and lncRNAs, and predicted network interactions.

The paper is written fairly well but would benefit with revisions to grammar and clarity.

The authors appropriately introduce fibrotic pathology generally and state, importantly, that there are commonalities amongst different fibrotic etiologies (e.g. the myofibroblast paradigm). The authors should expand more on this concept and better provide more information on fibrosis, how myofibroblasts contribute to fibrosis, the properties of myofibroblasts (including expression of specific genes), etc., which will be important for a reader's interpretation of this manuscript in the greater context of the literature. In addition to the cellular effects of FGF2 discussed by the authors in the introduction, the authors should also talk a little bit about the clinical effects demonstrated by FGF2 (Fiblast and trafermin are its other names) for burn wounds, incisions, skin grafts, etc., which have manifested prominently in the Chinese and Japanese literature in particular. In addition to the Dolivo et al. review in 2017 cited by the authors, see also See Nunes et al 2016 review in this journal titled "Fibroblast growth factors as tissue repair and regeneration therapeutics" for an excellent collection of much of the recent available data that they can incorporate into this manuscript introduction.

The authors should give a little bit of background on FGF2 and FGFs more generally. It's important to know that FGF2 is part of a superfamily of growth factors, that's it's part of a subfamily (FGF1), that is has 5 described isoforms but only one of which is typically secreted, that it acts via FGFR/MAPK pathways canonically, that it activates only certain receptor variants of particular FGFRs (see ornitz et al in JBC 1996).

Where is the FGF2 coming from? Is it purchased? Is it recombinant human FGF2? Was it produced recombinantly in the lab? What is the expression system? What is the purity? etc.

Recent literature (especially with the advent of single cell RNA-seq) has demonstrated extensive transcriptional and operative heterogeneity in fibroblast populations from different patients, from different areas of the skin, from different depths of skin, etc. There are far too many critical papers here to cite them all, but a good place to start would be Shaw and Rognoni's 2020 review article in Current Rheumatology Reports entitled "Dissecting fibroblast heterogeneity in health and fibrotic disease." As such, in order for the data generated in this manuscript to be useful, the authors needs to provide much more information about the patients from which these fibroblast samples were derived. We know that transcriptional differences lead to differences in, for example, receptor expression. Therefore, differences in gene expression may cause very real differences in the effects and magnitude of effects downstream of FGF2 signaling. How old were the patients? Where was the skin taken from? How many donors? etc. were fibroblasts pooled from different donors? This information all needs to be included.

The authors don't actually describe the stimulation with FGF2 in the methods section. Was the stimulation with FGF2 performed in the presence of FBS in culture? (This is fairly atypical and can make a big difference in gene expression changes). Figure 1a shows skin fibroblasts cultured for 48 hours at several indicated doses, but it's not obvious which dose was utilized for the samples harvested for RNA-seq. This information should be in the methods and in the results section and is very important. The authors should also justify the concentration that they used.

Also for the methods section, was DNA enzymatically degraded after Trizol extraction? Were any quality control steps performed? (e.g. RNA quality/degradation cutoff by RIN is common for transcriptomic analyses?)

This is not the first transcriptomic analysis published on FGF2 effects on human fibroblasts. How does this differ from previous analyses? (See Kashpur et al 2013 for example in BMC Genomics). The authors should make a point to focus on expression networks and lncRNAs here.

Picky comment but the authors state "Meanwhile, there were 213 LncRNAs with a fold change >2, among which 80 lncRNAs were up-regulated and 133 were down-regulated." It might be more clear to describe this as 213 lncRNAs with an absolute value(Fold change)>2.

The authors constructed lncRNA--mRNA expression networks. Many readers may not be familiar with this concept and how this prediction can be derived from RNA-seq data. The authors should do a much clearer job describing how this network is constructed, what the data mean, and what kind of predictions can be made from such data. As a researcher, what kind of testable hypotheses can I make from such a network? etc.

In the discussion authors state "Despite advancements in the understanding of the mechanism of skin wound repair, an effective method for accelerating the process remains to be justified." This statement is 1. too general (not always true, for example, as some growth factors can enhance wound healing under certain circumstances; this has been demonstrated in some scenarios in the clinic with FGF2 and PDGFBB), and 2. Not focused on the same fundamental principle as was set out in the introduction, which was focused around the possible uses of FGF2 as an antifibrotic, in keeping with extensive clinical (and preclinical) data. The authors should stick to the story about FGF2 as a negative regulator of fibrosis, as this is where the data are the strongest.

The authors don't give any information about the tissue culture substrate in the methods section, so this information must be included. Authors should acknowledge and discuss the fact that fibroblasts grown on tissue culture plastic or glass are subject to very high stiffnesses, well beyond what they would experience in a physiological situation (particularly in a homeostatic, healthy tissue). Therefore, this must be kept in mind when discussing FGF2-induced transcriptomic changes. Pathologically stiff fibroblasts tend to maintain myofibroblast phenotypes, so effects of FGF2 on in vitro tissue culture-grown fibroblasts may be more representative of the effects of pharmacologic exogenous FGF2 on myofibroblasts (as are the case with antifibrotic therapeutics) compared to the effects of endogenous FGF2 on quiescent fibroblasts in the papillary dermis, for example. It's an important distinction and the authors should acknowledge and address it.

The authors should compare their findings of specific genes being upregulated or downregulated with those of other reports looking at comparable FGF2-induced transcriptomic changes in fibroblasts, in order to lend greater credability to their results (as well as to the results of the papers that they cite!). Quick perusal of the supplementary excel file of DEGs and some simple find commands, and I find that the myofibroblast markers previously reported to be negatively regulated by FGF2 (for example ACTA2, CNN1, TAGLN, etc.) are reported to be downregulated here. That is a good sign! The authors should be more explicit about comparing their data to other published data to the extend that it makes sense/is practical to do so.

The authors should discuss lncs in general more, as well as the specific findings here regarding lncs.

Figure 1a should have scale bars.

Reviewer 3 ·

Basic reporting

An important concern is the use of English language is poor and ambiguous. The authors should re-check and revise carefully. Some examples of grammatical errors or typos are as follows (even in the abstract):
- FGF2 plays important roles in the skin wounds.
- A total of 1475 DEGs was identified.
- The current study comprehensive analyzed the FGF2-responsive transcriptional profile ...
- ...

The authors have not mentioned some related works related to bioinformatics analysis in FGF2 in skin fibroblasts and discuss about the differences between previous works and their work.

Experimental design

The authors used R language to perform bioinformatics analysis. It is important if the authors could show the source codes as supplementary information to help reproduce the results.

GO database or analysis has been used in previous works in bioinformatics such as PMID: 31277574 and PMID: 31921391. Therefore, the authors should refer more works in this description.

Validity of the findings

The authors should have some validation cohorts to support the findings.

Why did the authors select the cut-off value of KEGG pathway analysis is 7 pathways?

The authors should add more explanation in the figure's legends, now it did not contain enough information. For example, Fig. 1 is only the top-7 KEGG pathways, not all.

Additional comments

No comment.

---

## Round 0.2 · Minor Revisions

Kindly address the minor concerns raised by the reviewers.

Reviewer 1 ·

Basic reporting

Good

Experimental design

Good

Validity of the findings

They have validated RNAseq data using qPCR

Additional comments

Authors have incorporated all the suggested changes. Manuscript looks good and is ready to proceed further.

·

Basic reporting

None

Experimental design

None

Validity of the findings

None

Additional comments

The manuscript is much improved and I thank the authors for their incorporation of the critiques made by myself and the other reviewers. The manuscript in its current form is very good and the only changes I would recommend are regarding readability.

There still remain some awkward language issues at times. For example, "myofibroblasts are larger in area and have its structural characterization" should rather be "their structural..."., "and the fibronectin isoform" should read "and the ED-A fibronectin isoform" etc.

Minor comments:

The city that proteintech is based out of is called "Rosemont" not "Rosemount".

The authors say that FGF2 activates FGFR1 IIIb, and has low affinity for FGFR2 IIIb and FGFR3 IIIb, but that's not really correct. Check table 3 in the Ornitz JBC 1996 reference. According to this analysis, FGF2 activates FGFR1b, FGFR1c, FGFR2c, FGFR3c, and FGFR4.

The authors say "Importantly, FGF2 was observed to induce a shift in gene
expression to a more anti-fibrotic signature attenuated the expression of pro-fibrotic genes,
including collagen I, collagen III, α-SMA, and MMP-1(Dolivo et al. 2017b)"
The word "attenuate" means to decrease; Dolivo et al. 2017b (and other papers) demonstrate that FGF2 actually increases expression of MMP1 (which might be expected if FGF2 is antifibrotic, as MMP-1 degrades type I collagen) .

Reviewer 3 ·

Basic reporting

English language still needs to be improved.

In the previous comments, I suggested the authors to publish the R source code, but it has not been included in this version.

Experimental design

No comment

Validity of the findings

No comment

Additional comments

No comment

---

## Round 0.3 · accepted · Accept

The current revisions to the manuscript appear to have appropriately addressed the reviewer comments - Congratulations!